# An Optimized Protocol for In Vitro Regeneration of *Ocimum basilicum* cv. FT Italiko

**Sara Barberini** [1], **Chiara Forti** [2], **Marina Laura** [3], **Roberto Ciorba** [4], **Carlo Mascarello** [3], **Annalisa Giovannini** [3], **Barbara Ruffoni** [3] and **Marco Savona** [3,*]

[1]    Institute for Sustainable Plant Protection (IPSP CNR), Via Madonna del Piano 10, 50019 Sesto Fiorentino, Italy; sara.barberini@ipsp.cnr.it

[2]    Institute of Agricultural Biology and Biotechnology (IBBA CNR), Via Bassini 12, 20133 Milano, Italy; chiara.forti@ibba.cnr.it

[3]    CREA Research Centre for Vegetable and Ornamental Crops (CREA OF), Corso degli Inglesi 508, 18038 Sanremo, Italy; marina.laura@crea.gov.it (M.L.); carlo.mascarello@crea.gov.it (C.M.); annalisa.giovannini@crea.gov.it (A.G.); barbara.ruffoni@crea.gov.it (B.R.)

[4]    CREA Research Centre for Olive, Fruit and Citrus Crops, Via di Fioranello 52, 00134 Rome, Italy; roberto.ciorba@crea.gov.it

*    Correspondence: marco.savona@crea.gov.it; Tel.: +39-01-8469-4827

**Abstract:** Sweet basil (*Ocimum basilicum* L.; Fam. Lamiaceae) is an annual herbaceous plant with a high economic value used in folk medicine, pharmacology, and food production. In Italy, most of the varieties are used to produce the famous "pesto" sauce; however, almost all of them are susceptible to basil downy mildew (BDM) disease, strongly decreasing the growth of the fresh leaves and the survival of the whole plant. Nowadays, CRISPR/Cas9 technology is recognized to be a prominent way to enhance basil genetic breeding. In this work, we present an optimized protocol for in vitro direct regeneration of an elite cultivar, which is the major limiting factor for the transformation of *O. basilicum*. Regeneration has been obtained from different explants (leaves, cotyledons, cotyledonary nodes); the highest frequency has been obtained from cotyledonary nodes of seedlings germinated on MS medium containing TDZ. This protocol may be used for biotechnological applications as genome editing techniques to obtain basil-downy-mildew-disease-resistant clones.

**Keywords:** sweet basil; "pesto" sauce; cotyledonary nodes; direct regeneration



## 1. Introduction

*Ocimum* genus includes over 150 species, across which the species *basilicum* is spread throughout the Mediterranean and used as a culinary herb. Sweet basil (*Ocimum basilicum* L., Fam. Lamiaceae) is an annual herbaceous plant with high economic value, used in folk medicine [1], pharmacology [2], and food production due to its peculiar aroma and to its high antioxidant value [3]. A wide variety is available on the market, especially in morphology and essential oil composition (chemotypes, i.e., similar morphology but a different composition of the secondary metabolites of essential oil). Among "Genovese" types, which are characterized by a large leaf surface and by an essential oil rich in linalool and poor in estragol, FT Italiko is a cultivar commercially used in Italy for the production of the Genovese DOP (Protected Designation of Origin) "pesto" sauce, the most sold vegetable sauce all around the world [4]. Sweet basil is currently threatened by a severe fungal pathology caused by the oomycete *Peronospora belbharii* causing basil downy mildew (BDM) disease. Outbreaks of this disease was first described in Switzerland [5] and later throughout the world. In Italy, BDM has been reported since 2004 [6], causing a loss of production of all the most used cultivars in Italy [7]. To overcome this problem, various programs of genetic improvement have been conducted and, to-date, only a few cultivars are available on the market, combining a moderate resistance to the disease with valuable aromatic profiles [8]. The study of environmentally friendly control strategies presented alternative

measures to manage this problem, albeit with scarce success, at least in Italy [7]. In the last decade, new control strategies using BDM-resistant varieties have been proposed for horticultural crops [9] as the use of genome editing (GE) techniques, such as CRISPR/Cas9, to precisely inactivate the susceptibility (S) genes; however, the success in obtaining edited plants depends on a high efficiency of the ex novo regeneration of in vitro buds.

According to the literature, some protocols have been applied to the in vitro regeneration of sweet basil to be used in GE approaches; however, most of them are callus-mediated (indirect in vitro regeneration) and reported a high cultivar or even genotype-specific response [10,11].

Considering these aspects, we focused our attention on the elite cultivar *O. basilicum* FT Italiko with the aim to set up and optimize a suitable protocol for plant in vitro morphogenesis and direct regeneration, to be further used for CRISPR/Cas9 editing. Our previous preliminary results on the different varieties of *O. basilicum* [12] suggested that leaves and cotyledon are the best explants type to obtain promising percentages of in vitro regeneration; hence, we tested different explants as starting materials in media containing various plant growth regulators (PGR) combinations. As a result, we proposed a protocol using cotyledonary nodes as the best starting explant in a medium with thidiazuron (TDZ).

## 2. Materials and Methods

### 2.1. Plant Material, In Vitro and In Vivo Germination

In vitro: seeds of *Ocimum basilicum* L. 'FT Italiko' (La Semiorto Sementi®) were soaked for 20 min in a sodium hypochlorite solution (2.5% active chlorine) for surface-sterilization, and then rinsed three times with sterile water. Seeds were plated on Murashige and Skoog [13] medium supplemented with 8 g/L agar, at pH 5.7. The germination process was conducted in a growth chamber at $23 \pm 1$ °C in the dark for the first 3 days, then transferred under a photoperiod of 16 h of light (PPFD 30 $\mu$E m$^{-2}$ s$^{-1}$) and 8 h of darkness.

As starting materials for direct in vitro organogenesis experiments, hypocotyls and cotyledons from 2 weeks old seedlings and the first true leaves and cotyledonary nodes from 4 weeks old seedlings were considered.

In vivo: seeds were placed to germinate in a greenhouse on Klasmann® substrate and sterilized sand (70:30 *v/v*) and watered regularly. After one month, when the first true leaves were well-developed and extended, they were surface sterilized according to the protocol proposed by [14], in order to obtain sterile leaves explants to set up in vitro regeneration experiments.

### 2.2. Establishment of Cultures for Direct Regeneration

First true leaves of 4 weeks old seedlings, both reached by in vitro seedlings and by sterilized in vivo leaves, were used as starting explants and cut into small pieces as reported by [14], removing the midrib and the tip of the leaf. Explants were placed abaxial side up onto two different media, based on MSS shoot multiplication medium B [15] (MSS1 and MSS2, see Table 1 and Figure 1a) and to the same media in which two antioxidant compounds were added (MSS1aox and MSS2aox, see Table 1 and Figure 1a), i.e., citric acid (10 mg/L) and ascorbic acid (100 mg/L) to reduce the browning of the explants. Five Petri dishes with 10 explants for each medium were used, in three replicates (150 total explants/medium for MSS1 and MSS2) or in two replicates (100 total explants/medium for MSS1aox and MSS2aox).

Hypocotyl segments (about 5 mm long) of 2 weeks old seedlings of *O. basilicum* FT Italiko were excised and placed onto Petri dishes on three different media (MSS3, MSS2, or MSS2-aox, see Table 1 and Figure 1b) following the protocols of [16] and [14]. Explants were observed every 15 days and sub-cultured monthly for up 2 months. Four Petri dishes with 10 explants for each medium were used in three replicates (120 total explants/medium).

Cotyledons without petioles of 2 weeks old seedlings were placed onto a regeneration medium (MSS2 or MSS3, see Table 1 and Figure 1c); four Petri dishes with 20 explants for each medium were used in three replicates (240 total explants/medium).

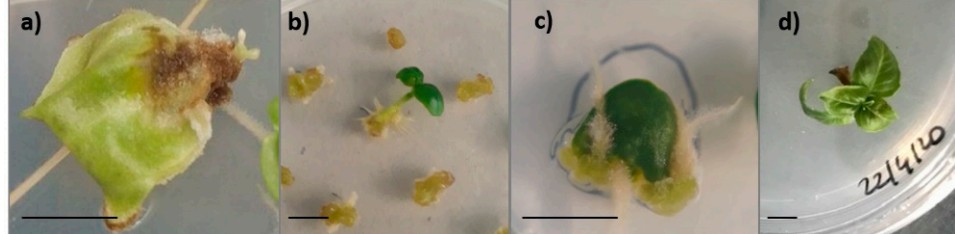

**Figure 1.** *Ocimum basilicum* FT Italiko: (**a**) leaf cultured on MSS1; (**b**) hypocotyls cultured on MSS3; (**c**) cotyledons cultured on MSS2; (**d**) CN cultured on MSS2. (MSS = shoot multiplication medium B; CN = cotyledonary node). Scale bar = 0.5 cm.

**Table 1.** Starting plant materials and media composition used in the regeneration experiments. Basal medium MSS with 3% sucrose has been used as control.

| Explant Type | PGR (Concentration) Added to MSS Medium | Compounds (Concentration) Added to MSS Medium | Medium Acronym |
|---|---|---|---|
| First true leaf (in vitro or in vivo) | BA (1 mg/L) | / | MSS1 |
| | BA (1 mg/L) | +Citric acid (10 mg/L) + Ascorbic acid (100 mg/L) | MSS1aox |
| | TDZ (4 mg/L) | / | MSS2 |
| | TDZ (4 mg/L) | +Citric acid (10 mg/L) + Ascorbic acid (100 mg/L) | MSS2aox |
| Hypocotyls (in vitro) | IAA (1 mg/L) | +ZnSO$_4$ (12.9 mg/L) | MSS3 |
| | TDZ (4 mg/L) | / | MSS2 |
| | TDZ (4 mg/L) | +Citric acid (10 mg/L) + Ascorbic acid (100 mg/L) | MSS2aox |
| Cotyledons (in vitro) | IAA (1 mg/L) | +ZnSO$_4$ (12.9 mg/L) | MSS3 |
| | TDZ (4 mg/L) | / | MSS2 |
| Cotyledonary nodes (CNs) (in vitro) | TDZ (4 mg/L) | / | MSS2 |

Cotyledonary nodes (CNs), consisted in a piece of about 0.8 cm in length within the epicotyl, the hypocotyl, and the petioles of the cotyledons, were excised from 4 weeks old seedlings. All the explants were placed onto a regeneration medium (MSS2, see Table 1 and Figure 1d). Three Petri dishes with 15 explants/medium were used in three replicates (135 total explants/medium).

The explants materials of all the experiments were placed in a growing chamber at 23 ± 1 °C, with a photoperiod of 16 h of light (PPFD 30 μE m$^{-2}$ s$^{-1}$) for hypocotyls, cotyledons, and CNs, while in the dark for leaves. Explants grown on media supplemented with TDZ were first placed in the dark for 15 days, as reported by [14].

For all the experiments, hormone-free MS shoot multiplication medium B [15] with 3% sucrose was used as control (4 Petri dishes per experiment).

### 2.3. Data Collection and Statistical Analysis

Each experiment was performed at least two or three times, as described in Section 2.2. The shoot regeneration percentage (%), defined as the number of explants with de novo regenerated shoots on total explants considered, and the mean number of shoots per explant were recorded after each subculture. Maximum score refers to the highest mean value collected among the experiments. Collected data were statistically analyzed using a computer software (Statistica 10®). Experimental results were subjected to an analysis of variance (Kruskall-Wallis ANOVA or unpaired *t*-test in case of a comparison of 2 groups) with appropriate post-hoc tests. Percentages were converted in angular values before statistical analysis. Results are presented as mean ± standard error (SE).

## 3. Results and Discussion

Plant regeneration, which is the major limiting factor for a transformation protocol in *O. basilicum*, has been obtained for only a few specific cultivars [14,17]. It is known that genotype is the factor that influences the development of regeneration systems, particularly for a species with numerous cultivars such as sweet basil; for "genovese-type" sweet basil, the most commonly used in the food industry, a few protocols are available that are standardized exclusively for this specific cultivar [11,18].

In our work, we proposed a regeneration protocol for the elite cultivar *O. basilicum* FT Italiko, commonly used in Liguria (Italy) for the production of the DOP "pesto" sauce. All the experiments were conducted starting from the same batch of seeds to avoid (as much as possible) any intra-varietal genotypic variations.

With regard to the leaf explants, it is possible to conclude that the addition of cytokinins, in particular BA (at 1 mg/L), induces positive effects on the regeneration capacity (Figure 2). In fact, for FT Italiko, it was possible to reach a maximum regeneration percentage (32.6%) from in vivo first true leaves (see Table 2) compared with the 36.8% reported by [14] for green purple ruffles cultivar. Macroscopically, it was seen that, after two subcultures, the margins of the leaves showed symptoms of browning, thus decreasing the regenerative capacity. The experiment was then repeated and implemented with the addition of antioxidants to the culture medium MSS1: the addition of citric and ascorbic acids resulted in slightly better results in terms of regeneration percentage, although not statistically significant ($p = 0.165$, unpaired *t*-test, $n = 20$). In addition, all explants also showed an evident aptitude for callogenesis at the cut point, suggesting the possibility to follow an alternative method, i.e., indirect regeneration process.

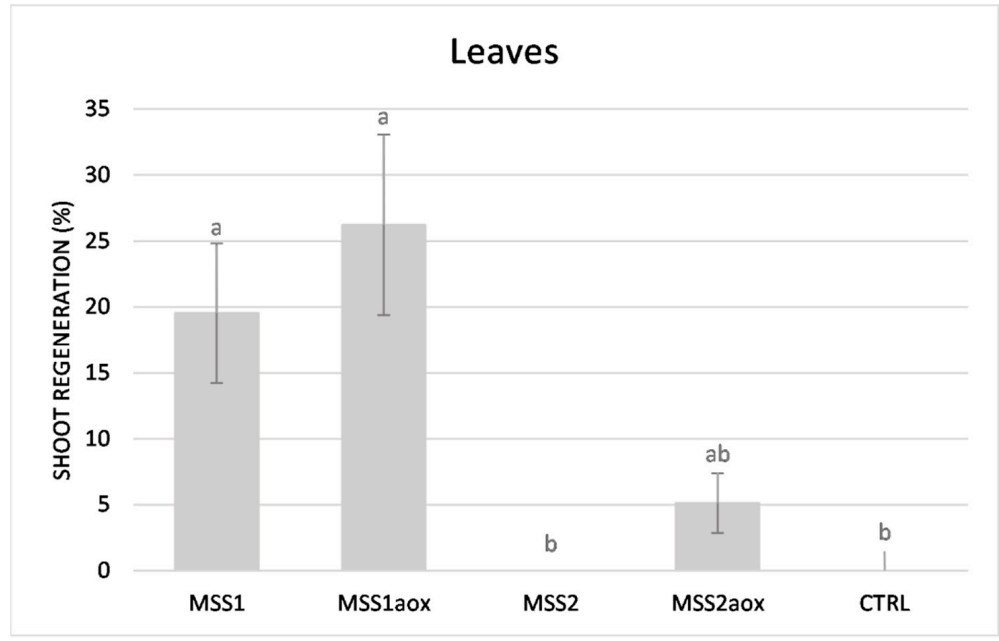

**Figure 2.** Influence of the hormones and antioxidants on the regeneration aptitude observed from in vivo and in vitro first true leaves as starting materials (data recorded after 2 months). Error bars represent standard error. Different letters indicate significant differences between treatments at $p < 0.05$ (Tukey HSD test's results). For culture media acronym, refer to Table 1 (MSS1: shoot multiplication medium B + 1 mg/L BA; MSS1aox: shoot multiplication medium B + 1 mg/L BA + citric acid (10 mg/L) + ascorbic acid (100 mg/L); MSS2: shoot multiplication medium B + 4 mg/L TDZ; MSS2aox: shoot multiplication medium B + 4 mg/L TDZ + citric acid (10 mg/L) + ascorbic acid (100 mg/L); CTRL: control, MSS PGR-free).

Regeneration percentages obtained from hypocotyls or from cotyledons did not provide appreciable results either, with very low regeneration percentages of de novo shoots

(with a maximum of 9.18% and 19.7% of regeneration on MSS2ox medium for hypocotyls and for cotyledons, respectively, see Table 2 and in Supplementary Table S1). A small amount of callus was observed only on the cutting surface of the explants but direct organogenesis was recorded as regenerated shoots arise from explant tissue but not from callus. Rhizogenesis was observed on cotyledon explants, while hypocotyls showed browning damages after 30 days. Our results on the elite cultivar are in contrast with those proposed by Verna et al. [16], in which a simple protocol for a commercial variety of sweet basil regeneration has been proposed with the use of zinc sulphate; this observation suggests and confirms a strong different genotype dependence.

**Table 2.** Shoot regeneration (%) and mean n. of shoots per responsive explants: maximum score recorded among the experiments related to different explant types. Data are presented as mean $\pm$ SE.

| Explant Type | Medium | Shoot Regeneration (%) | Mean n. of Shoots per Responsive Explants |
|---|---|---|---|
| Leaves (in vivo) | MSS1aox | 32.64 $\pm$ 7.79 | 1.52 $\pm$ 0.131 |
| Hypocotyls | MSS2aox | 9.18 $\pm$ 2.43 | 1.33 $\pm$ 0.333 |
| Cotyledons | MSS2aox | 19.7 $\pm$ 8.31 | 1.25 $\pm$ 0.214 |
| CNs | MSS2 | 93.5 $\pm$ 2.31 | 2.6 $\pm$ 0.254 |

Finally, observing that some shoots arose from the basal part of cotyledons close to the junction of the node, we evaluated the possibility of direct regeneration from cotyledonary nodes, as was also reported in *O. gratissimum* by Khan and collaborators [19]. The use of CN as the starting explant provided excellent results (Table 2): more than 93% of explants grown on medium containing TDZ showed direct regeneration, without intermediary callus phases, with an average of 2.6 shoots per explant produced. The regenerated plants showed normal morphological and growth characteristics, showing a complete whole plant regenerated in vitro after 30 days. Furthermore, all the regenerated shoots of sweet basil were able to root spontaneously in 40 days onto MSS medium PGR-free (Figure 3). The regeneration of a whole plant from modified cells and tissues, through in vitro culture techniques, represents the main bottleneck for the application of genome editing in different plant species. In the frame of the Italian project "Biotech-GEO", aiming at developing sweet basil cultivars resistant to basil downy mildew, the development of a suitable and fast protocol for direct plant regeneration is of primary importance. Currently, this protocol has been used in an experimental trial with the use of CRISPR/Cas9 technology in sweet basil by our research group (Laura et al., 2021) [20].

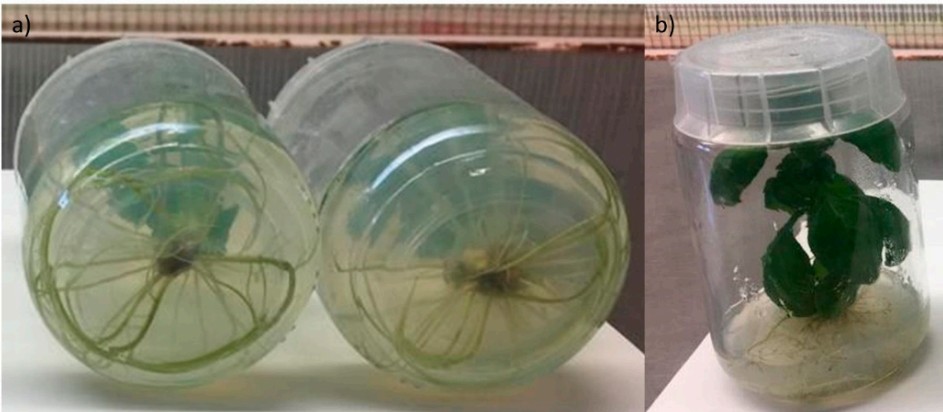

**Figure 3.** *Ocimum basilicum* FT Italiko: (**a**) rooting phase of regenerated plants onto MSS PGR-free; (**b**) example of singularized and rooted plants onto MSS PGR-free (MSS = shoot multiplication medium B).

## 4. Conclusions

In this study, we established a complete and reliable protocol for plant in vitro regeneration of *Ocimum basilicum* FT Italiko, an elite cultivar of sweet basil used in North Italy to produce the famous "pesto" sauce. Cotyledonary nodes cultured on MSS+TDZ (4 mg/L) was the best somatic tissue type to reach a high regeneration percentage. Obtaining in vitro plant from this cultivar is strongly important for the following and future studies in the era of genome editing, since it is essential to reach a good regeneration protocol prior to establishing transformation and editing experiments. Most of the sweet basil varieties used for culinary aims (and in Italian cuisine) are in fact susceptible to the basil downy mildew disease; this problem has drastically compromised basil crops in Italy, with very severe damages to the farmers. The promising results on the availability of a suitable regeneration system for the cultivar of interest is a fundamental prerequisite that creates the possibility to improve the resistance to *P. belbharii* via CRISPR/Cas9 technique.

**Supplementary Materials:** The following supporting information can be downloaded at: https://www.mdpi.com/article/10.3390/horticulturae9030407/s1. Table S1: Influence of PGR and compounds on the shoot regeneration (%) on different starting explant types.

**Author Contributions:** Conceptualization, M.S., B.R., S.B., A.G. and M.L.; methodology, S.B., R.C., C.F., M.L., C.M. and M.S.; statistical analysis, S.B.; writing—original draft preparation, S.B.; writing—review and editing, S.B., M.S., C.F., M.L. and B.R.; project administration and funding acquisition, M.S. All authors have read and agreed to the published version of the manuscript.

**Funding:** The funding support of the Italian Ministry of Agriculture, Food and Forestry Policies (MiPAAF), re-named in 2022 as "Ministero dell'agricoltura, della sovranità alimentare e delle foreste" (Masaf)—Project "Biotech" (grant number D.M. n. 15947 del 18/05/18) is acknowledged.

**Data Availability Statement:** The data that support the findings of this study are available on request from the corresponding author Marco Savona, upon reasonable request.

**Conflicts of Interest:** The authors declare no conflict of interest.

## Abbreviations

| | |
|---|---|
| BA | 6-benzyladenine |
| IAA | indole-3-acetic acid |
| DOP | Denominazione di Origine Protetta (Protected Designation of Origin) |
| MS | medium Murashige and Skoog (1962) medium |
| MSS | Murashige and Miller shoot multiplication medium B |
| PGR | plant growth regulator |
| TDZ | thidiazuron |

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
