# Peer review of "An Optimized Protocol for In Vitro Regeneration of Ocimum basilicum cv. FT Italiko"

_horticulturae, doi:10.3390/horticulturae9030407_

Round 1

Reviewer 1 Report

The paper aimed at optimizing protocol for in vitro regeneration of elite basil cultivar. However, the presented results do not show complete regenerants (rooted plants), while the experimental setup and the presentation of the results are occasionally unclear and raise many questions. The paper needs to be rewritten.  

To make this paper publishable, the authors need to respond to the suggested corrections and comments to the text, listed below:

In Material and Methods section (Lines 113-114), it is stated that “Frequency (%) of explants developing shoots and mean number of shoots per explant were recorded after each subculture.”, but there is no mention of shoot regeneration percentage. What is shoot regeneration percentage? Is it the percentage of responsive explants, or – since number of shoots per explant is not presented for any explant/medium type, anywhere in the text – is it the mean number of shoots per responsive explant / % of responsive explants? The authors need to define shoot regeneration percentage. If it is just the percentage of responsive explants, then the number of shoots per responsive explant must be presented as well.

The maximum score presented in Table 2 should also be defined in Material and Methods section.

Most importantly, the regeneration frequencies / the percentage of responsive explants, and number of shoots per responsive explant / should be presented for each explant type and each treatment, as means ± SE, and with statistical significance of those findings – not only the maximum score, for some of the explants and only some of the media, without showing which homogenous groups they belong to.

Material and Methods

When selecting the optimal explant type, why wasn’t the regeneration response of each explant type tested for each treatment (nutrient media)?

Lines 76-77:  “they were surface sterilized according to the protocol proposed by [14]” and  Lines 81-83: “First true leaves of 4 weeks old seedlings, both reached by in vitro seedlings and by sterilized in vivo leaves were used as starting explants and cut in small pieces as reported by [14], removing the midrib and the tip of the leaf.”

The paper cited here as the reference 14 (Phippen, W.B.; Simon, J.E. Anthocyanins in basil (Ocimum basilicum L.). Journal of Agricultural and Food Chemistry, 1998, 46(5), 240 1734-1738. DOI: 10.1021/JF970887R) is not dealing with in vitro culture of basil, hence there is no mention of surface sterilization of basil leaves in it, nor does it describe leaf explants used for in vitro culture establishment. It was probably cited erroneously and this must be corrected in the manuscript.

The above also applies to Lines 90-92: “Hypocotyl segments (about 5 mm long) of 2 weeks old seedlings of O. basilicum FT Italiko were excised and placed onto Petri dishes on three different media (MSS3, MSS2 or MSS2-aox, see Table 1 and Figure 1b) following the protocols of [15] and [14]”, as well as to Lines 105-106: ”Explants grown on media supplemented with TDZ were first placed on dark for 15 days, as reported by [14].”

Lines 84, 85... 91; ... 198: What exactly is “Murashige and Miller Shoot Multiplication medium B”, abbreviated as MSS medium? Is it different from MS medium mentioned earlier in the text? In which case, either the differences should be clearly described or another reference (describing MSS medium) should be provided. Furthermore, when introducing MSS abbreviation in the text (Material and Methods section), the full name of the medium should be provided at first mention. 

Lines 99-100: What part of the cotyledons has been removed, and what was the size of it? When describing CN explants, it is not mentioned whether the roots were excised from 4 weeks old seedling. If they were excised, that should be stated too.

Line 102: 135 total explants per medium

Results and Discussion

Line 122: “CN regenerated on MSS2” should be CN cultured on MSS2 (or shoot regenerated on CN...)

Lines 124-127: “Ocimum basilicum is a horticultural species traditionally cultivated in open field, and it is reported that it could be also propagated in vitro for its medicinal value [16, 17]; however, very often a propagation through nodal explants or directly from seeds is erroneously described as regeneration [18, 19]”

This sentence is redundant, especially here in the Results and Discussion section. It should be removed.

As for plant regeneration, see reviews by Long et al. (2022, Frontiers in Plant Science; https://www.ncbi.nlm.nih.gov/pmc/articles/PMC9280033/pdf/fpls-13-926752.pdf), or Ikeuchi et  al. (2016; Development; https://journals.biologists.com/dev/article/143/9/1442/47902/Plant-regeneration-cellular-origins-and-molecular): “An entire plant can be regenerated from an adult tissue or organ, a mass of unorganized calli, or even a single cell in a process referred to as plant regeneration. Plant regeneration refers to the physiological renewal, repair, or replacement of tissue in plants (Ikeuchi et  al., 2016).” According to both, propagation by axillary buds/shoots (which further develop adventitious roots) is considered as a regeneration of the whole plant.

Line 138: “in particular auxin (BA, 1 mg/L)”

BA belongs to cytokinins, not auxins. Therefore, ‘auxin’ should be replaced with ‘cytokinin(s)’.

Lines 137-139: “As regards the leaf explants, it is possible to conclude that the addition of plant growth regulators, in particular auxin (BA, 1 mg/L) produced positive effects on the regeneration capacity (Figure 2).”

Since no PGRs other than cytokinins were used with leaf explants – hence we don’t know what potential effect other PGRs might exert, it would be far more appropriate if this sentence was reformulated as “As regards the leaf explants, it is possible to conclude that the addition of cytokinins, in particular BA (at 1 mg/L) produced positive effects on the regeneration capacity (Figure 2).”

Lines 140-141: “comparable with 36.8% reported by [14] for ‘Green Purple Ruffles’”

Again the reference, regarding the in vitro regeneration percentage, is made to a paper that does not deal with in vitro culture. This must be corrected.

Line 159: “very low regeneration percentages of direct shoots” – I suggest using the term other than direct: ‘de novo shoots’ or ‘adventitious shoots’

Line 168: What is ”the inner part of cotyledons”? Part of cotyledons ‘close to the junction of the node’ should be designated as ‘proximal’ or ‘basal’ part, rather than the ‘inner’ part

Figure 2, Table 2. Why are regeneration percentages in Figure 2 given for pooled leaf explants (in vitro and in vivo taken together), while in Table 2 maximum regeneration frequency is presented for in vivo leaves only? When trying to optimize the protocol for plant regeneration, where it is crucial to choose the right starting explant, it would be desirable to see whether there actually was some difference in the response of the respective leaf explants, i.e. to which extent they differed. The regeneration frequencies should be statistically analyzed and presented separately for in vitro and in vivo leaves.

Lines 173-175: The regenerated plants showed normal morphological and growth characteristics, showing a complete whole plant regenerated in vitro after 30 days.” Where is this shown?  

Conclusion

Lines 179-180: “a complete and reliable protocol for plant in vitro regeneration” - a complete protocol for plant regeneration should include obtaining the whole plant, i.e. the rooting step for adventitious shoots, which is not described in this work. That unless all the shoots rooted spontaneously, in which case this ought to be mentioned and documented.

Abbreviations

Lines 195 IBA indole-3-butyric acid; 198 MSS Murashige and Miller Shoot Multiplication medium B; 199 NAA naphthaleneacetic acid

IBA and NAA are not mentioned in the text at all; MSS is given as an abbreviation, but the full name of the medium is lacking, and so is its composition or reference to it.

Reviewer 2 Report

The communication titled:"An optimized protocol for in vitro regeneration of Ocimum basilicum cv. FT Italiko", submitted to Horticulturae falls perfectly within the scope of the journal. The authors suggest an optimized protocol of regeneration of sweet basil cv FT Italiko in vitro.

Overall, the paper provides a clear and concise overview of the topic. It effectively highlights the economic value of sweet basil and the challenges posed by BDM disease. The mention of CRISPR/Cas9 technology as a potential solution to this problem is also relevant and interesting. Yet, some major revisions must be considered before considering the paper for publication. 

Introduction: 

* It would be helpful to define some of the technical aspects and terms used in this section, such as "chemotype", "Genoverse DOP", among others, for readers who may not be familiar with these terms.

* While the introduction provides a good overview of the challenges faced by sweet basil, it could benefit from a more explicit statement of the research question or hypothesis that the study aims to address.

* There is no mention of the scientific basis of choosing cotyledonary nodes as the best starting explant, and how this choice relates to the goal of using CRISPR/Cas 9 editing. 

Materials and Methods: 

* On what basis the concentrations of plant growth regulators were chosen ? did the others conduct a preliminary study to find these concentrations ?

** Line 141: For ‘Green Purple Ruffles’ cultivar.

** Line 181: replace cotiledonary with cotyledonary.

The manuscript “An optimized protocol for in vitro regeneration of Ocimum basilicum cv. FT Italiko” (Manuscript ID: horticulturae-2279895) may be published after major corrections.

Reviewer 3 Report

In this short communication manuscript (horticulturae-2279895) entitled " An optimized protocol for in vitro regeneration of Ocimum basilicum cv. FT Italiko" submitted to Horticulturae, Sara Barberini and colleagues present an optimized protocol for in vitro direct regeneration of an elite cultivar, which is the major limiting factor for transformation of O. basilicum. Regeneration has been obtained from different explants (leaves, cotyledons, cotyledonary nodes): the highest frequency has been obtained from cotyledonary nodes of seedlings germinated on MS medium containing TDZ. This study is an interesting work, I have however several concerns that may be addressed to improve the quality of the work.

1. Authors emphasized the potential application of this in vitro regeneration protocol in the basil protection against downy mildew disease in the Abstract. However, there is no data in the results section to support this statement. Please discuss this potential in details in the revision.

2. For Figure 1, scale bar should be included in pictures, and full names for abbreviations FT, MSS and CN should be spelt out the revised legend.

3. For Figure 2, hormones and antioxidants employed in this assay should be laid out in the revised figure.

4. For Tables 1 and 2, the medium acronyms MSS1, MSS1aox, MSS2, MSS2aox, and MSS3 are confusing. Author should modify and explain these acronyms in details in the revision.

Round 2

Reviewer 1 Report

The authors responded to most of the corrections, comments and remarks, and the manuscript has been significantly improved, although they failed to show statistical significance of the results (homogenous groups are not shown in Table 2 or Supplementary Table S1).

I am aware that the paper has been proposed as a short communication, with the attention focused on the best results suitable for the research in progress, and although they found that MSS2 provides very high regeneration percentage, I still think that they might consider other media with CN explants in their future research – if not regarding the regeneration percentage, then maybe regarding the mean number of shoots per explant.  

Reviewer 2 Report

All of the reviewers' concerns have been satisfactorily addressed by the authors. The manuscript is suitable for publication in its current form.